# Perturb-and-max-product: Sampling and learning in discrete energy-based models

**Miguel Lázaro-Gredilla, Antoine Dedieu, Dileep George**
Vicarious AI
SF Bay Area, CA
{miguel, antoine, dileep}@vicarious.com

## Abstract

Perturb-and-MAP offers an elegant approach to approximately sample from an energy-based model (EBM) by computing the maximum-a-posteriori (MAP) configuration of a perturbed version of the model. Sampling in turn enables learning. However, this line of research has been hindered by the general intractability of the MAP computation. Very few works venture outside tractable models, and when they do, they use linear programming approaches, which as we show, have several limitations. In this work, we present perturb-and-max-product (PMP), a parallel and scalable mechanism for sampling and learning in discrete EBMs. Models can be arbitrary as long as they are built using tractable factors. We show that (a) for Ising models, PMP is orders of magnitude faster than Gibbs and Gibbs-with-Gradients (GWG) at learning and generating samples of similar or better quality; (b) PMP is able to learn and sample from RBMs; (c) in a large, entangled graphical model in which Gibbs and GWG fail to mix, PMP succeeds.

## 1 Introduction

In this work, we concern ourselves with the problem of black-box parameter estimation for a very general class of discrete energy-based models (EBMs), possibly with hidden variables. The golden measure of estimation quality is the likelihood of the parameters given the data, according to the model. However, the likelihood of EBMs is not computable in polynomial time due to the partition function, which in turn makes its optimization (parameter estimation) difficult.

This is an active research problem, with multiple approaches having been proposed over the last decades [37, 29, 33, 2, 13, 14, 12, 37, 17, 38, 19]. These methods typically fall into one or more of the following categories (a) they depart from the maximum likelihood (ML) criterion to obtain a tractable objective, but in doing so they fail when the data does not match the model (e.g., pseudolikelihood [13]); (b) they can only be applied to fully observed models (e.g., piecewise training [29]); (c) they fail catastrophically for simple models (e.g., Bethe approximation [12]); (d) they require additional designs (such as a custom encoder) for new models, i.e., they are not black-box (e.g, Advil [19]).

A general approach to discrete EBM learning that does not fall in any of those pitfalls is sampling: as we review in Section 2.1, sampling from an EBM results in efficient learning. Exact sampling is unfeasible in general, but approximate sampling works well enough in multiple models. Gibbs sampling is the most popular because it is simple, general, and can be applied in a black-box manner: the sampler can be generated directly from the model, even automatically [5]. Its main limitation is that for some models, mixing can be too slow to be usable. Recently the Gibbs-with-gradients (GWG) [8] sampler was proposed, which significantly improves mixing speed, enabling tractability of more models. A completely different approach to learning is Perturb-and-MAP [22], but this approach has not seen widespread adoption because the general MAP problem could not be solved fast enough or with good enough quality, see Section 2.4.1.

35th Conference on Neural Information Processing Systems (NeurIPS 2021).

In this work, we show that max-product, a relatively under-utilized algorithm, can be used to perform good enough MAP inference in many non-tractable cases very fast. Thus, perturb-and-max-product not only expands the applicability of perturb-and-MAP to new scenarios (we show that it can sample in highly entangled and multimodal EBMs in which the state-of-the-art GWG and traditional Perturb-and-MAP fail) but can also be much faster in practice than the state-of-the-art GWG.

## 2 Background

We review here ML learning, Perturb-and-MAP, belief propagation, and how they connect.

### 2.1 Sampling and learning in discrete EBMs

**Notation:** Consider an EBM with variables $x$ described by a set of $A$ factors $\{\phi_a(x_a; \theta_a) = \theta_a^\top \phi_a(x_a)\}_{a=1}^A$ and $I$ unary terms $\{\phi_i(x_i; \theta_i) = \theta_i^\top \phi_i(x_i)\}_{i=1}^I$: $x_a$ is the subset of variables used in factor $a$, $\theta_a$ is a vector of factor parameters and $\phi_a(x_a; \theta_a)$ is a vector of factor sufficient statistics. For the unary terms, the sufficient statistics $\phi_i(x_i; \theta_i)$ are the indicator functions, which are set to 1 when the variable takes a given value and to zero otherwise. The energy of the model can be expressed as $E(x) = -\sum_{a=1}^A \phi_a(x_a; \theta_a) - \sum_{i=1}^I \phi_i(x_i; \theta_i)$ or, collecting the parameters and sufficient statistics in corresponding vectors, $E(x) = -\Theta_a^\top \Phi_a(x) - \Theta_i^\top \Phi_i(x) = -\Theta^\top \Phi(x)$.

Given a discrete data set $\mathcal{D} \equiv \{x^{(n)}\}_{i=1}^N$, where $x \in \{0, \ldots, M-1\}^D$, the log-likelihood of the parameters for a generic EBM in which some variables ($x_h$) are hidden can be expressed as

$$\mathcal{L}(\Theta|\mathcal{D}) = \frac{1}{N} \sum_{n=1}^N \log p(x^{(n)}; \Theta) = \frac{1}{N} \sum_{n=1}^N \log \sum_{x_h} \exp(\Theta^\top \Phi(x^{(n)}, x_h)) - \log \sum_x \exp(\Theta^\top \Phi(x)) \quad (1)$$

The learning problem can be described as efficiently maximizing the key quantity Eq. (1). However, Eq. (1) can only be computed for small models, due to the summation over $x$ (the partition function), which involves an exponential number of summands. In order to use gradient ascent methods, we need the gradient of Eq. (1) w.r.t. $\Theta$, which is $\nabla_\Theta \mathcal{L}(\Theta|\mathcal{D}) = \frac{1}{N} \sum_{n=1}^N \langle \Phi(x^{(n)}, x_h) \rangle_{p(x_h|x^{(n)};\Theta)} - \langle \Phi(x) \rangle_{p(x;\Theta)}$. From this gradient we can make two standard observations: First, at a maximum of Eq. (1), the expectation of the sufficient statistics according to the data and the model match, since the gradient needs to be zero. Second, sampling enables efficient learning: we can efficiently approximate the expected sufficient statistics, with no bias, by sampling from the model (both unconditionally and conditioned on $x^{(n)}$). Exact sampling is also unfeasible, so we turn to approximate sampling.

### 2.2 Perturb-and-MAP

In [22], a family of samplers, including the following convenient form, was proposed:

$$x^{(s)} = \underset{x}{\arg\max} \left\{ \Theta_a^\top \Phi_a(x) + (\Theta_i + \varepsilon^{(s)})^\top \Phi_i(x) \right\} \quad (2)$$

where $\varepsilon^{(s)}$ is a random perturbation vector with as many entries as $\Theta_i$ (i.e., $\sum_i |x_i|$ entries, where $|x_i|$ is the cardinality of variable $x_i$). This means that to generate a sample, first we draw $\varepsilon^{(s)}$ from some distribution, use it to additively perturb the unary terms, and then we solve a standard maximum a posteriori (MAP) problem on the perturbed model.

While MAP inference is NP-complete, it is easier than partition function computation which is #P-complete. In fact, in log-supermodular models, such as attractive Ising models, the exact MAP can be computed using graph-cuts, but the partition function still needs exponential time.

The distribution of the samples $\{x^{(s)}\}$ will of course depend on the perturbation density. We exclusively use independent, zero-mean densities $\varepsilon \sim \text{Gumbel}(\varepsilon; -c, 1)$, where $c \approx 0.5772$ is the Euler-Mascheroni constant. With this choice, models with only unary terms result in the exact Gibbs distribution $p(x^{(s)}) \propto \exp(\Theta^\top \Phi(x^{(s)}))$ (see Lemma 1, [22]). More realistic models containing non-unary terms, will in general *not* match the Gibbs density, but will approximate it.

The Perturb-and-MAP approach also allows for the computation of an upper bound on the partition function, $\log \sum_x \exp(\Theta^\top \Phi(x)) \leq \langle \max_x(\Theta_a^\top \Phi_a(x) + (\Theta_i + \varepsilon)^\top \Phi_i(x)) \rangle_{p(\varepsilon)}$, (see Corollary 1,

[10]). If we use this approximation in the log-likelihood (1), we get a much easier target for optimization

$$\hat{\mathcal{L}}(\Theta|\mathcal{D}) = \frac{1}{N}\sum_{n=1}^{N}\left\langle \max_{x_h}(\Theta_a^\top \Phi_a(x^{(n)}, x_h) + (\Theta_i + \varepsilon)^\top \Phi_i(x^{(n)}, x_h))\right\rangle_{p(\varepsilon)}$$

$$- \left\langle \max_x(\Theta_a^\top \Phi_a(x) + (\Theta_i + \varepsilon)^\top \Phi_i(x))\right\rangle_{p(\varepsilon)}. \tag{3}$$

whose gradient is $\nabla_\Theta \hat{\mathcal{L}}(\Theta|\mathcal{D}) = \frac{1}{N}\sum_{n=1}^{N}\Phi(x^{(n)}, x_h)_{\hat{p}(x_h|x^{(n)};\Theta)} - \langle\Phi(x)\rangle_{\hat{p}(x;\Theta)}$, where $\hat{p}(\cdot;\Theta)$ is a (possibly conditional) sampler of the form of Eq. (2). Note that this gradient has the same form as the gradient of the true likelihood, but with the exact sampler replaced by the approximate one. So this can be thought of as the approximate gradient of the true log-likelihood, or as the true gradient of the approximate log-likelihood.

The expectations in Eq. (3) and its gradient can be easily approximated, without bias, using the sample mean. Thus, we can use stochastic gradient descent (SGD) and minibatches, to get a generally applicable technique for graphical models *if* we can solve the MAP problem efficiently. So far, the application of perturb-and-MAP has been restricted to models whose structure allowed exact MAP solutions or allowed LP relaxations of the MAP problem to produce accurate results. As we show in Section 5.2, even simple Ising models violate these assumptions.

## 2.3 The Bethe approximation and belief propagation

To make perturb-and-MAP more widely useful, we need a more general (even if approximate) approach to MAP inference. Here we describe a way to achieve this leveraging the Bethe free energy, which uses a variational formulation and a modified entropy to approximate the partition function.

Using an arbitrary variational distribution $q(x)$ we can lower bound the partition function using the expectation of the free energy. At the maximum with respect to this $q(x)$, we obtain the equality:

$$\beta^{-1}\log\sum_x \exp(\beta\Theta^\top\Phi(x)) = \max_{q(x)}\left\{\langle\Theta^\top\Phi(x)\rangle_{q(x)} + \beta^{-1}\mathcal{H}(q(x))\right\} \tag{4}$$

where $\mathcal{H}(q(x)) = -\sum_x q(x)\log q(x)$ is the entropy and $\beta$ is an *inverse temperature* parameter. At $\beta = 1$, the l.h.s. of Eq. (4) becomes the log-partition function. At $\beta \to \infty$ and with perturbing $\Theta$, the problem posed by Eq. (4) matches the one in Eq. (2). We keep this temperature parameter throughout our derivation. The maximization in (4) is still intractable, so we introduce two approximations and we obtain

$$\max_{q_a(x_a), q_i(x_i)} \sum_{a=1}^{A}\langle\phi_a(x_a;\theta_a)\rangle_{q_a(x_a)} + \sum_{i=1}^{I}\langle\phi_i(x_i;\theta_i)\rangle_{q_i(x_i)} + \beta^{-1}\mathcal{H}_B(q_a(x_a), q_i(x_i))$$

$$\text{s.t.} \qquad \sum_{x_{a\backslash i}}q_a(x_a) = q_i(x_i), \ \ \sum_{x_i}q_i(x_i) = 1, \ \ q_a(x_a) \geq 0,$$

which is the Bethe approximation of the log-partition function [39]. These approximations are (a) the use of the Bethe free entropy $\mathcal{H}_B(q_a(x_a), q_i(x_i)) = \sum_a \mathcal{H}(q_a(x_a)) + \sum_i (1 - f_i)\mathcal{H}(q_i(x_i))$ (which is exact for trees, approximate for loopy models) instead of the exact entropy; and (b) the use of locally consistent pseudo-marginals $q_a(x_a)$ and $q_i(x_i)$, which might not correspond to any joint density $q(x)$.

To convert this into an unconstrained optimization problem, we move to the dual. The dual variables are called "messages", since there is one of them associated to each factor-variable pair. See [40] for details. Setting the gradient to 0, we recover the belief propagation fixed-point equations

$$m_{i\to a}(x_i) = \phi_i(x_i;\theta_i) + \sum_{b\in \text{nb}(i)\backslash a} m_{b\to i}(x_i)$$

$$m_{a\to i}(x_i) = \beta^{-1}\log\sum_{x_{k\backslash i}} e^{\beta\left(\phi_a(x_a;\theta_a) + \sum_{k\in\text{nb}(a)\backslash i}m_{k\to a}(x_k)\right)},$$

where $\text{nb}(\cdot)$ denotes the neighbors of a factor or variable. Updates must proceed sequentially, factor by factor. Convergence to a fixed point is not guaranteed for loopy models. We recover the primal quantities from the messages. In particular, the unary marginals are $q_i(x_i) \propto \exp(\beta(\phi_i(x_i;\theta_i) + \sum_{b\in\text{nb}(i)}m_{b\to i}(x_i))$. The recovered primal value in Eq. (5), for $\beta = 1$, is an approximation of the log-partition function, but cannot be used for learning in general, see [23, 12], and Section 5.1.

## 2.4 Max-product message passing

We are interested in the limit $\beta \to \infty$ (zero temperature). The l.h.s. of Eq. (4), i.e. the quantity that we are approximating, becomes $\max_x \Theta^\top \Phi(x)$, exactly the MAP problem. The update for $m_{i \to a}(x_i)$ remains unchanged, the other simplifies to $m_{a \to i}(x_i) = \max_{x_{k \setminus i}} \phi_a(x_a; \theta_a) + \sum_{k \in \text{nb}(a) \setminus i} m_{k \to a}(x_k)$.

The new "unary marginals" $x_i = \arg\max_c \left\{ \phi_i(x_i = c; \theta_i) + \sum_{b \in \text{nb}(i)} m_{b \to i}(x_i = c) \right\}$ provide the maximizing configuration. This algorithm is known as max-product or belief revision. A damping factor $0 < \alpha \le 1$ in the message updates can be used without affecting the fixed points but improving convergence. In fact, even parallel updates often provide satisfactory results with as little as $\alpha = \frac{1}{2}$.

### 2.4.1 The connection with linear programming

When taking the limit $\beta \to \infty$, the astute reader will have probably noticed that, rather than modifying the fixed point updates, we could have removed the entropy in the r.h.s. of Eq. (5) and obtained a simple linear program (LP). Indeed, this is possible, and there is a vast literature on how to tackle the resulting LP (a continuous relaxation of the MAP problem) [7, 21, 16, 35, 20, 15, 34]. In fact, most approximate MAP inference research falls in this category, exploiting the convenience of the convexity of LPs.

It is common, although incorrect, to then assume that taking the limit $\beta \to \infty$ in the fixed point equations, i.e., max-product, is just some sort of coordinate-wise dual algorithm to optimize the same LP. This is not the case. The connection between the LP and max-product has been studied in detail in [36]. Max-product is a minimax algorithm in the dual, and has different fixed points than the LP.

Here we argue that, for the purpose of sampling, max-product is superior to LP-based solvers. The perturbed MAP problems are highly multimodal, even for simple cases such as Ising models. The Bethe free energy is a non-convex function with multiple minima. As $\beta \to \infty$, all those minima flatten out, but still exist [36], and become the fixed points of max-product, so the multimodality of the problem is preserved, and hopefully one of the modes is reached. In contrast, the LP approach merges all those minima into a single point or flat region, the global minimum, and the structure is lost. The relaxed solution of the LP on simple Ising models is often 0.5 for all or most variables, a result that cannot be rounded to any useful discrete answer. We assess this in our experiments.

## 3 Perturb-and-max-product (PMP) for sampling and learning

As explained in Section 2.4.1, LP-based solvers are not well-suited for perturb-and-MAP inference, and fail even in simple cases. Furthermore, max-product is an anytime algorithm whose precision can be controlled by the number of times that the messages are updated, much like the update steps of Gibbs sampling, enabling an easy trade-off between compute time and accuracy.

Perturb-and-max-product (PMP) learning is presented in Algorithm 1. Minibatched PMP samples are used to estimate the gradient of the approximate likelihood Eq. (3). The procedure is highly parallel: at each time step we compute the updates for all the factors (and for all the samples in the minibatch) at the same time. All the computation is local in the graph: updates for messages *from* a factor or variable only involve messages *to* that factor or variable. In Algorithm 1, we use $\Theta|_{x^{(n)}}$ to refer to the modification[1] of the unary terms in $\Theta$ that clamps the visible variables to the observed values $x^{(n)}$. The damping $\alpha$ is set to $\frac{1}{2}$, since this value seems to offer a good trade-off between accuracy and speed in most cases.

A natural benchmark for this general-purpose algorithm is Gibbs sampling (GS). Both GS and PMP have a computational complexity that scales as $\mathcal{O}(T \sum_{i=1}^I f_i)$, where $f_i$ is the number of factors variable $i$ is connected to, and $T$ is the number of times we do a full sweep over all the variables. Sequentially, PMP is a constant factor slower than GS, given the more involved updates. However, the high parallelism of PMP makes it up to orders of magnitude faster in practice. Recently, a strong competitor to GS was presented: Gibbs with gradients (GWG) [8]. GWG essentially optimizes the

---

[1] Observe that if for visible variables $x_i$ we modify the unary parameters $\theta_i$ so as to attribute 0 score to the visible choice $x_i = x_i^{(n)}$ and $-\infty$ score to any other choice, the problem of finding the MAP for that new $\Theta|_{x^{(n)}}$ is effectively the same as finding the MAP for the original $\Theta$, but clamping the visible variables to $x^{(n)}$.

order in which variables in the model are updated, thus providing faster mixing for the same number of updates (and with 2-3× times more compute), so we also consider it in our comparisons.

Intuitively, Gibbs sampling propagates less information than PMP, since rather than real-valued messages, it only propagates discrete values. To illustrate this effect, consider a simple chain EBM: Gibbs sampling can be exponentially slow [1], whereas PMP finds the optimal configuration in linear time. The trade-off is that PMP is not unbiased (even asymptotically) and a bit less general than GS: it can only be applied when all the factors in the model are max-marginalizable (the max-product updates for the factor are tractable). Fortunately, most small factors and some not-so-small ones[2] are, and they can be combined without limitation into rich EBMs. Neural functions are typically not easy to max-marginalize, and therefore using a whole neural network (NN) to define the energy is not compatible with PMP. A NN that is broken into max-marginalizable pieces, however, can be used.

One important distinction when using PMP for learning is not to confuse the learned $\hat{\Theta}_{\text{PMP}}$ (which are the parameters that the PMP sampler uses to try to match to the data distribution) with the parameters of a Gibbs distribution. As we show in Section 5.1, the learned $\hat{\Theta}_{\text{PMP}}$ can be very far from the Gibbs distribution parameterization that produces the same density.

---

**Algorithm 1:** Learning and sampling with perturb-and-max-product (PMP)

---

**Input:** Sufficient statistic $\Phi(\cdot)$, data $\{x^{(n)}\}_{n=1}^{N}$, step size $\eta$, minibatch size $S$, sampling steps $T$
**Output:** A learned set of parameters $\Theta$
$\Theta \sim \mathcal{N}(0, 0.01)$                                               // Init
**for** $l \leftarrow 1$ **to** $L$ **do**
    **for** $s \leftarrow 1$ **to** $S$ **do**
        $n \leftarrow \text{Uniform}(1, N)$                          // Choose a sample from the training set
        $y_+^{(s)} \leftarrow \text{pmp}\left(\Theta|_{x^{(n)}}, T\right)$                    // Get a sample from the posterior
        $y_-^{(s)} \leftarrow \text{pmp}\left(\Theta, T\right)$                          // Get a sample from the prior
    **end**
    $\Theta \leftarrow \Theta + \eta\left(\frac{1}{S}\sum_{s=1}^{S}\Phi(y_+^{(s)}) - \frac{1}{S}\sum_{s=1}^{S}\Phi(y_-^{(s)})\right)$          // Stochastic gradient ascent
**end**
**return** $\Theta$
**Procedure** pmp($\Theta$, $T$)
    $\varepsilon_i \sim \text{Gumbel}(-c, 1)\ \forall i;\ \ m_{i\to a}^{(0)}(x_i) \leftarrow 0\ \forall a, i, x_i;\ \ m_{a\to i}^{(0)}(x_i) \leftarrow 0\ \forall a, i, x_i$          // Init
    **for** $t \leftarrow 1$ **to** $T$ **do**
        $m_{i\to a}^{(t)}(x_i) \leftarrow \phi_i(x_i; \theta_i + \varepsilon_i) + \sum_{b\in\text{nb}(i)\setminus a} m_{b\to i}^{(t-1)}(x_i)\ \forall a, i, x_i$          // Max-product
        $m_{a\to i}^{(t)}(x_i) \leftarrow \frac{1}{2}m_{a\to i}^{(t-1)}(x_i) + \frac{1}{2}\max_{x_{k\setminus i}}\left(\phi_a(x_a; \theta_a) + \sum_{k\in\text{nb}(a)\setminus i} m_{k\to a}^{(t)}(x_k)\right)\ \forall a, i, x_i$
    **end**
    $x_i \leftarrow \arg\max_c\left\{\phi_i(x_i = c; \theta_i + \varepsilon_i) + \sum_{b\in\text{nb}(i)} m_{b\to i}(x_i = c)\right\}\ \forall i$          // Decode
    **return** $x$

---

## 4  Related work

Learning using Perturb-and-MAP has been proposed in different variants in recent literature, [22, 10, 6, 26, 25, 11, 25], with the common theme being that either only tractable MAP inference was considered or LP solvers (including graph-cuts in this category) were used. As we argue in Section 2.4.1, this approach does not work on multimodal problems (such as those invariant to permutations like in Section 5.6) and performs poorly even in simple settings, as we show in Sections 5.2 and 5.3.

In the seminal work [22], the authors suggest using QPBO [24] for the MAP problem, which, as a convex method, also fails even in small Ising models. The approaches in [10, 11] use the max-product LP (MPLP) [7] to solve the MAP problem, whose poor performance in non-attractive problems we show in Section 5.2. The recent work [25] is limited to tractable models. The structured learning of [26] also relies on log-supermodular potentials that make MAP inference tractable.

---

[2]For instance, n-wise factors require $\mathcal{O}(C^n)$ compute (where $C$ is the cardinality of the variables), logical factors (such as the OR of $n$ variables) only need $\mathcal{O}(n)$ time [20], cardinality potentials limiting $k$ out of $n$ variables to be ON need $\mathcal{O}(kn)$ [30], other higher-order potentials accept efficient computation [31], etc.

A different approach is proposed in [6]: it absorbs the parameters of the model in the perturbation distribution, and projects each data point to a concrete perturbation value (the MAP of the perturbation given the data point). The parameters of the perturbation are then slightly adjusted towards fitting the projected data, and the process is repeated, in the form of hard expectation maximization. The optimizations are "inverse" problems in the continuous space of perturbations, typically solved using quadratic programming. This approach has the advantage of being able to handle any noise distribution, but it is very computationally demanding and does not support Gumbel-distributed perturbations.

## 5 Experiments

We now study the empirical performance of PMP for sampling and learning. We measure performance using the maximum mean discrepancy (MMD) [9] between a set of model samples $x$ and a set of ground truth samples $x'$:

$$\text{MMD}^2(x, x') = \frac{1}{NN'} \sum_{i=1}^{N} \sum_{j=1}^{N'} k(x^{(i)}, x'^{(j)}) + k(x^{(i)}, x'^{(j)}) - 2k(x^{(i)}, x'^{(j)})$$

where $k(x^{(i)}, x'^{(j)}) = \exp - \left( \frac{1}{D} \sum_{d=1}^{D} [x_d^{(i)} \neq x_d'^{(j)}] \right)$ is the standard average Hamming distance kernel. MMD is also used in the original GWG paper [8]. Annealed importance sampling, while popular, is inadequate for this use case, see Appendix B.4 for details.

We use the term "full sweep" to refer to a full MCMC update for all variables in GS and GWG, and a single update of all the messages for PMP.

Our experiments are run on a GeForce RTX 2080 Ti. All compared methods are coded on JAX [4] and run with changes only in the samplers for maximum comparability of their runtimes. Code to reproduce these experiments can be found at `https://github.com/vicariousinc/perturb_and_max_product/`. The specific max-product updates used in the presented models are derived in Appendix C. Code automating max-product inference in general models can be found in [41].

### 5.1 Learning and sampling from the "wrong" model

We want to clarify that during learning PMP selects the parameters $\hat{\Theta}_{\text{PMP}}$ so as to match the expected sufficient statistics of the data, when *sampling*. Those are *not* the parameters of a Gibbs distribution. To drive this point home, and also to show that PMP can learn and sample from a distribution that cannot be learned with the Bethe approximation, we use the toy model from [23].

Consider an EBM with 4 variables, fully connected, with energy function $E(x) = -\theta \sum_{i<j \in \text{edge}} x_i x_j$, where $x \in \{-1, +1\}^4$. This model is not learnable using the Bethe approximation for $\theta > 0.316$ [23]. We set $\theta = 0.5$ and run PMP learning on infinite data sampled from this model (this is feasible, since the expected statistics can be computed exactly). We use Adam for 200 iterations with 0.01 learning rate: each iteration considers 100 chains and run 100 full sweeps. The result is $\hat{\theta}_{\text{PMP}} \approx 0.331$. One can be tempted to compare this value with the original $\theta = 0.5$ of the generating Gibbs distribution to assess the quality of the PMP approximation, but that would be incorrect. $\hat{\theta}_{\text{PMP}}$ is the parameter of the PMP sampler, not of a Gibbs distribution. To see this, note that the KL divergence from data to (a) the Gibbs distribution $\text{KL}(p(x|\theta = 0.5) \ || \ p(x|\hat{\theta}_{\text{PMP}} = 0.331)) = 0.119$, is much worse than to (b) the learned sampling distribution $\text{KL}(p(x|\theta = 0.5) \ || \ p_{\text{PMP}}(x|\hat{\theta}_{\text{PMP}} = 0.331)) \approx 0.008$, which produces an almost perfect match. One way to look at this is that we are learning the "wrong" model, but it is wrong in such a way that it compensates for how PMP sampling is also inexact, thus resulting in a learned sampling distribution that is closer to that of the data.

### 5.2 MPLP: Max-product is not broken

In the seminal paper "Fixing max-product" [7], Globerson et al. show how the LP in Eq. (5) (for $\beta \to \infty$) can be solved via a convergent sequence of message passing updates, which they call max-product LP (MPLP). We show that MPLP can perform very poorly even in simple cases. We

first consider a 2D cyclical square lattice with energy $E(x) = -0.1 \sum_{i,j \in \text{edge}} x_i x_j$, and side sizes $\{5, 10, 15\}$. We then generate 100 fully-connected Ising model with dimensions $\{5, 10, 15\}$ and energy $E(x) = -\sum_{i,j \in \text{edge}} w_{ij} x_i x_j - \sum_i b_i x_i$ where $w_{ij}$ is uniformly sampled in $[-2, 2]$ and $b_i$ in $[-0.1, 0.1]$, with $x \in \{-1, +1\}$. For both problems, we calculate the exact partition function and estimations using PMP and MPLP (see Section 3). We run PMP for 200 full sweeps, and MPLP until the message updates change by less than $10^{-6}$ (which corresponds to more than 200 full sweeps in practice). The same perturbations are used for both methods, so they both solve the same exact MAP problem. The estimations returned are upper bounds, so the approximation error should be positive. In Figure 1[left] we see that this is the case for the 2D lattice, where both approaches perform similarly. However, for Ising models, neither method provides an actual upper bound (since the MAP inference and the expectation are only computed approximately), but MPLP is significantly worse, see Figure 1[center]. Furthermore, MPLP is a sequential algorithm (for which the damping trick used to turn max-product into a parallel algorithm does not work in practice): for the $15 \times 15$ 2D lattice sampling from MPLP uses $500\times$ more compute time than PMP, making it impractical for sampling.

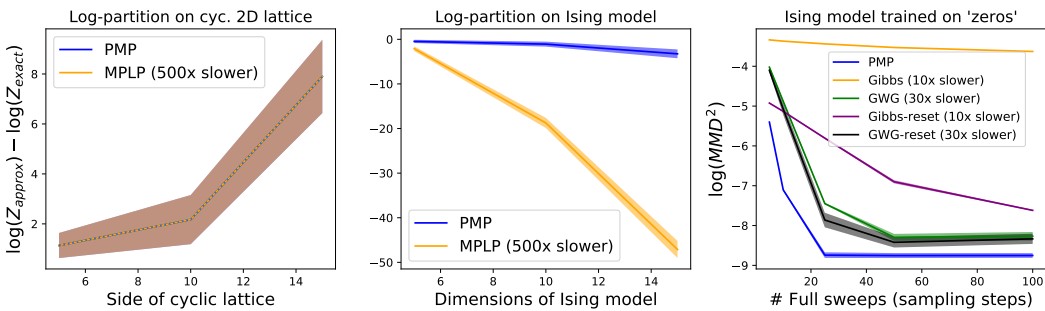

Figure 1: [Left and center] Log-partition estimation error using PMP and MPLP. Closer to zero is better. Shaded area represents $\pm 1$ std deviation. [Right] See section 5.3. Sampling quality for different methods, keeping training fixed and varying the number of full sweeps at sampling time.

## 5.3 Ising model: MNIST zeros

In this experiment, we train a fully connected Ising model on a dataset of contours derived from the 0s of the MNIST dataset [18] (see top row of Figure 4[bottom left]). For learning, we initialize all the parameters to 0 and use Adam for 1000 iterations with 0.001 learning rate. At every iteration, we consider 100 chains and run 50 full sweeps. The MCMC chains are persistent, and a non-persistent version which resets on every learning iteration is marked with the suffix "reset". After training, we produce new samples following each model and method for a varying number of sampling steps (see Figure 1[right]).

As expected, GWG outperforms Gibbs. The reset versions are truncated, biased samplers, during learning, so they perform better when sampling for a similar number of full sweeps. Despite the prevalence of persistent contrastive divergence (PCD), short-run reset samplers seem best when we want to generate a high quality sample with a truncated MCMC chain. PMP outperforms all the other methods, and does so with a much smaller compute time, as indicated in the legend. PMP is actually performing around $3\times$ more operations than Gibbs, but its parallelism more than makes up for it.

We also use an LP solver (instead of max-product) with Pertub-and-MAP to learn the Ising model (see Appendix D). This approach reaches a $\log(\text{MMD}^2)$ of $-4.33(\pm 0.20)$. LP performance is poor and the approach is unpractical: it took three orders of magnitude more compute. See the LP samples at the bottom row of Figure 4[bottom, left] and how they compare with the other methods.

## 5.4 Structured models: 2D lattices and Erdös-Renyi graphs

Here we reproduce the experiments from [8] with structured models, following precisely the definition, sampling, and learning described. See details in Appendix B. The authors used the root-mean-square error (RMSE) of the estimated parameters (which we include for reference) as their performance

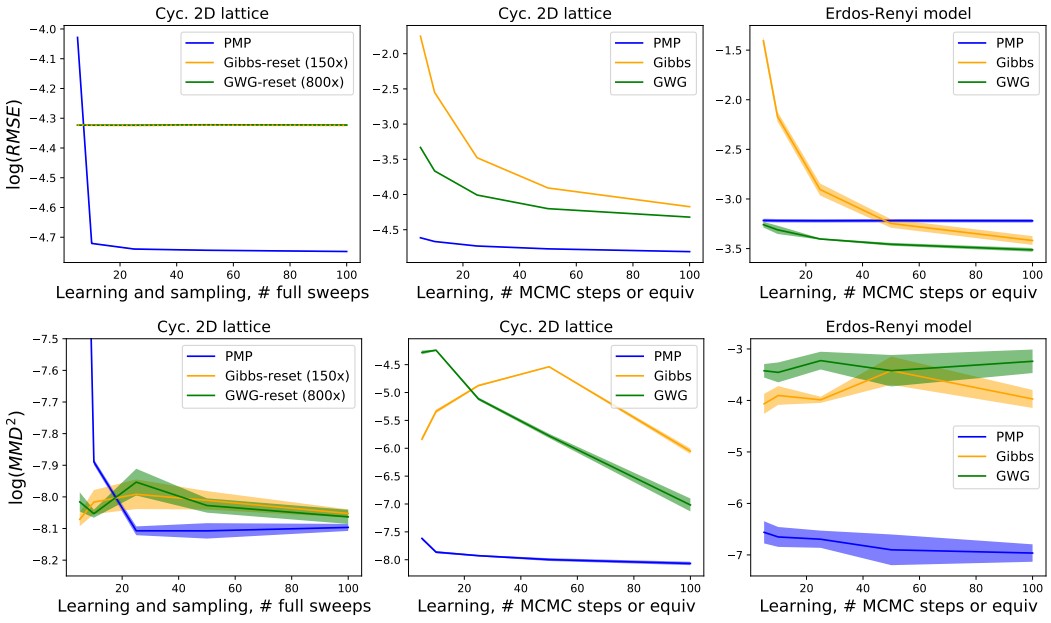

Figure 2: Performance on the lattice model ($\theta = -0.1$, side 25) and the Erdös-Renyi graphs from [8]. [Left column] x-axis determines the number of full sweeps for training and sampling, for all methods. [Center and right columns] x-axis determines the number of *individual variable updates* for Gibbs and GWG at training time. PMP is trained for a number of full sweeps that result in less training time than GWG at the same x-axis location. To sample, all methods are run for 50 full sweeps.

metric, but this metric is not as meaningful for PMP, since as we saw in Section 5.1, a large difference in the parameters does not imply low quality samples.

**Non-persistent training:** In the left column of Figure 2 we show the performance of training in a 2D lattice from [8] with 20 chains and a varying number of full sweeps both for learning and sampling. The chains are reset in between learning iterations. All methods perform well, with PMP taking the lead (even in RMSE), and also being substantially faster (see legend).

**Persistent training:** This is the actual setting from [8], shown in the remaining plots of Figure 2. Less than a full sweep is run in between iterations and chains are not reset. Samples are produced by a fixed 50 full sweeps for all methods. The x-axis measures here the number of *individual variable updates* in between learning iterations, so all the points in the x-axis correspond to less than a full sweep. A persistent version of PMP is outside the scope of this work (although we provide a possible approach in Appendix A). Just as the non-persistent MCMC samplers, non-persistent PMP cannot produce meaningful samples in less than a full sweep, so it cannot compete by that metric. Instead, we adjust the number of full sweeps of PMP to take equal or smaller compute time than GWG at the same x-axis location. This allows a few full sweeps of PMP, sufficient to significantly outperform GWG with the same compute budget. Also, notice that once again the persistent regime is not the most conducive to the highest sampling quality for any of the methods tested. Hence, despite being faster and more commonly used, this regime might not be the best choice for all applications.

## 5.5 Restricted Boltzmann machines: learning and sampling

Here we train a restricted Boltzmann machine (RBM) on MNIST data and sample from it using PMP. Its energy is defined as $E(x_v, x_h) = -\sum_{i,j} w_{ij} x_{vi} x_{hj} - \sum_j b_j x_{hj} - \sum_i c_i x_{vi}$, with $x_v \in \{0,1\}^{N_v}, x_h \in \{0,1\}^{N_h}$. To the best of our knowledge, PMP is the first reported instance of a perturb-and-MAP method able to train an RBM and sample recognizable MNIST-like digits from it. Indeed, we ran the author's code from [32] and the samples did not look like MNIST digits by a far margin. Note that no samples are reported in that work. See Appendix B.4 for discussion.

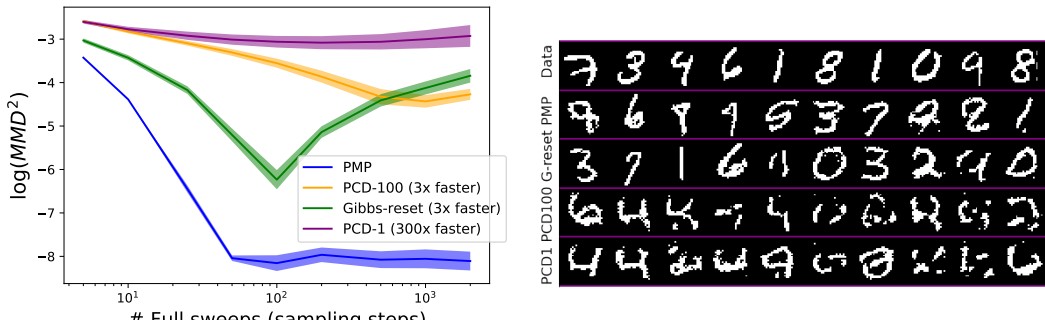

Figure 3: [Left] Sampling quality for an RBM trained once and sampled for an increasing number of full sweeps. [Right] Samples per method at 100 full sweeps. See Appendix B.2 for more samples.

We train an RBM with 500 hidden units on MNIST using 100 non-persistent full sweeps of PMP and block Gibbs (Gibbs-reset) and also single-sweep (PCD-1) and 100-sweep (PCD-100) persistent block Gibbs variants. We initialize $W \sim \mathcal{N}(0, 0.01)$ and $b, c \sim \mathcal{N}(0, 1)$. We use stochastic gradient descent with learning rate $0.01$ and train for $200$ epochs using minibatches of size $100$.

Note that here it does not make sense to use GWG, since the block Gibbs sampler is much more efficient than a black-box selection of the next variable to sample. The block Gibbs sampler is informed by the structure of the model. However, PMP is not. It is processing all factors in parallel, unaware of the conditional independence in the RBM.

After learning, the MMD for each model as the number of full sweeps increases is reported in Figure 3[left]. It is interesting to see the marked peak of Gibbs-reset at the 100 full sweeps mark. Indeed, we have learned a biased sampler tuned for 100 full sweeps, so more MCMC steps make that model *worse*. Again, PMP outperforms the competition in sample quality. However, in this particular case, the block Gibbs sampler specifically designed for RBMs can leverage as much parallelism as PMP, and having a smaller multiplicative constant due to its simplicity, it is faster than PMP. The trade-off is not being black-box and providing worse MMD figures.

Figure 3[right] shows sampling results for each method after 100 full sweeps. Samples for other numbers of full sweeps and a brief discussion can be found in Appendix B.2. Similar to Section 5.3, we tried to use an LP solver instead of max-product. We were able to get it to work to some extent on a smaller dataset, as reported in Appendix B.3, but the LP approach was still outperformed by PMP.

### 5.6 2D blind deconvolution

Finally, we consider the problem of sampling from an EBM with $40680$ binary variables that are intricately connected, and which are invariant to permutations.

A binary 2D convolution generative model combines two sets of binary variables $W$, and $S$ to form a set of binary images $X$. We illustrate in the top row of Figure 4 for the generation of 2 independent images. Each binary entry of $W$ and $S$ is modeled with an independent Bernoulli prior. The fact that the pixels in $W$ are contiguous and form lines is not captured by the model, and is done only to produce shapes that humans can parse more easily.

$W$ (size: $n_{\text{feat}} \times \text{feat}_{\text{height}} \times \text{feat}_{\text{width}}$) contains several 2D binary features. $S$ (size: $n_{\text{images}} \times n_{\text{feat}} \times h \times w$) is a set of binary indicator variables representing whether each feature is present at each possible image location. The two are combined by convolution, placing the features defined by $W$ at the locations specified by $S$ in order to form an image. The image is binary, so feature overlap acts as an OR, not a sum. The example in the top row of Figure 4 shows how the features $W$ on the left are placed at the locations marked by $S$, with the top row of $S$ producing the top image in $X$ and the bottom row of $S$ producing the bottom image in $X$. Each column of $S$ corresponds to the locations of each of the 5 available features in $W$.

Observe that, unlike $S$, $W$ is shared by all images. This makes all variables highly interdependent and the posterior highly multimodal: permuting the first dimension of $W$ and the second dimension of $S$ in the same manner does not change $X$, so this naturally results in multiple equivalent modes. The

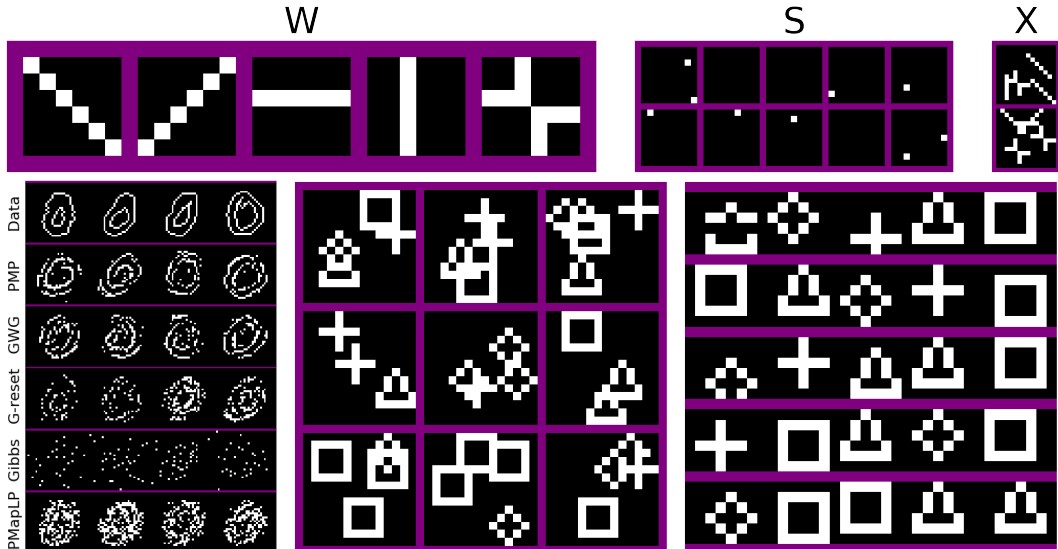

Figure 4: Top row: Binary 2D convolution example, with features ($W$), locations ($S$) and resulting images ($X$). Bottom row: [Left] Data and samples from MNIST zeros Ising model after 25 full sweeps of sampling using each tested method. [Center] 9 examples of blind deconvolution dataset (full dataset $X$ in the Appendix B). [Right] 5 samples of the posterior using PMP.

proposed problem is a Bayesian version[3] of the task in [27], where the weights have been converted into binary variables. Note that Boolean matrix factorization emerges as a particular case.

Since this model has no free parameters, we do not need to train it. Instead, we directly turn to inference: given $X$ (which contains 100 images generated according to a binary 2D convolution), we sample from the joint posterior $p(W, S|X)$. Out of those 100 images, 9 are shown in Figure 4[bottom row, center]. On its right, 5 samples (one per row) from the posterior are obtained by PMP after 1000 full sweeps. Each sample comes from a consecutive random seed, showing that the model is consistently good at finding samples from the posterior. The dimensions of $W$ used for the generation of $X$ were $4 \times 5 \times 5$, but during inference they were set to $5 \times 6 \times 6$ to simulate a more realistic scenario in which we do not know their ground truth values. The samples from the posterior used that additional feature to include an extra copy of the symbols used for generation, but otherwise looked correct. The size of $S$ was set to $100 \times 5 \times 9 \times 9$, so that the convolution output matches the size of $X$. GWG was much slower per full sweep, and we were not able to produce any meaningful posterior samples in a reasonable time. GWG did work when $S$ or $W$ were clamped to the true values, but not when trying to sample from both simultaneously. All 100 images in $X$ are provided in Appendix B.

## 6  Discussion

In this work, we present PMP, a method for sampling and learning in EBMs. PMP improves over standard, LP-based Perturb-and-MAP in that it offers a MAP solver that is fast, parallel, anytime, and produces high quality solutions in problems where LP-based MAP solvers fail (e.g., simple non-attractive Ising models and challenging models with symmetries, such as the one in Section 5.6).

Compared with Gibbs sampling and GWG, PMP offers faster mixing for a given number of full sweeps, and its parallelism helps it complete those full sweeps in substantially less time. PMP also has the ability to solve some problems exponentially faster than Gibbs sampling, both theoretically in the case of chains and experimentally in cases like Section 5.6. Its main limitation is that it is only applicable in EBMs consisting of max-marginalizable factors, although this class is very rich, contains many well-known models, and can be made arbitrarily complex.

---

[3]A different way to pose this problem is to assume that $W$ is a set of parameters of the generative model, and infer them via maximum likelihood. Because we place a prior on them and infer their posterior instead, we are doing full Bayesian inference.

## Acknowledgments and Disclosure of Funding

We thank Wolfgang Lehrach and Guangyao Zhou for useful discussions during the preparation of this manuscript.

This work was supported by Vicarious AI and ONR grant N00014-19-1-2368.

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
