# Appendices

## A    Persistent PMP

During learning, persistent Gibbs sampling chains allow obtaining samples that are close to the target density in very few steps (typically only one) by warm-starting the sampling from the samples at the previous learning iteration. The idea is that since the weight update is small, the target distribution cannot have changed too much, and thus very few sampling steps are needed to take the Gibbs chain back to convergence at the new equilibrium distribution. This is the basis of persistent contrastive divergence, which significantly speeds up training when compared with bringing Gibbs sampling to convergence from scratch at each learning iteration.

PMP as presented in this work is not persistent, so it does not enjoy this benefit, although it is still very fast (but note that the lack of persistency might result in higher quality samples, as shown in Section 5). Here we want to show how we can make PMP compatible with persistency during learning. In principle, at each learning iteration we use a brand new perturbation, which means that the MAP problem is completely new and the solutions from previous steps cannot be leveraged to accelerate the current iteration. However, rather than creating an independent perturbation at each learning iteration, we could "perturb the previous perturbation" to obtain a very similar one. If both the perturbation and the weights are very similar to the previous ones, then the MAP problem as a whole is also very similar to the previous one, and we can bootstrap it from the solution obtained in the previous learning iteration.

As to the question of how to generate correlated variables with marginal Gumbel densities, a simple solution is to generate a chain of correlated normal variables following the marginal density $\mathcal{N}(0, 1)$ and then transform them to follow the Gumbel density:

$$\delta^{(t)} \sim \mathcal{N}(0, 1) \ \forall t$$
$$\gamma^{(0)} \sim \mathcal{N}(0, 1)$$
$$\gamma^{(t)} = \sqrt{\rho}\gamma^{(t-1)} + \sqrt{1 - \rho}\delta^{(t-1)}$$
$$\varepsilon^{(t)} = \mathrm{Gumbel}_{\mathrm{CDF}}^{-1}(\mathrm{Normal}_{\mathrm{CDF}}(\gamma^{(t)})) \ \forall t.$$

Here, all the $\gamma^{(t)}$ variables follow a Gaussian density of mean 0 and variance 1, with a degree of correlation controlled by $0 \leq \rho \leq 1$. For $\rho = 0$ they are independent, whereas for $\rho = 1$, they all take the same value. The same is true for $\varepsilon^{(t)}$, but they will be marginally distributed as Gumbel. Using a large enough $\rho$ and a small enough $\eta$ (the learning rate), the MAP problem barely changes between learning iterations and it can be warm-started from the solution of the previous iteration.

## B    Experimental details

### B.1    Details of the structured models experiments in Section 5.4

The experiments of this section follow exactly the details from [8], some of which are taken from the paper and some of which are taken from the code.

About the Erdös-Renyi model:

- It consists of 200 binary variables randomly connected so that the average number of neighbors of a node is 4.
- The energy expression for this model is $E(x) = x^\top W x + b^\top x$, where $W$ is symmetric and has zero diagonal. The binary variables $x$ are in $\{-1, +1\}^{200}$. Note that the customary 0.5 factor on the pairwise weights is missing, so each weight is counted twice.
- When two nodes are connected, the weight of their connection is drawn from $\mathcal{N}(0, 0.25)$ (the weight will be counted twice, see the previous item).
- There is a constant bias term in all the nodes of 0.1 (this is known from the authors' code).

About the cyclic lattice model:

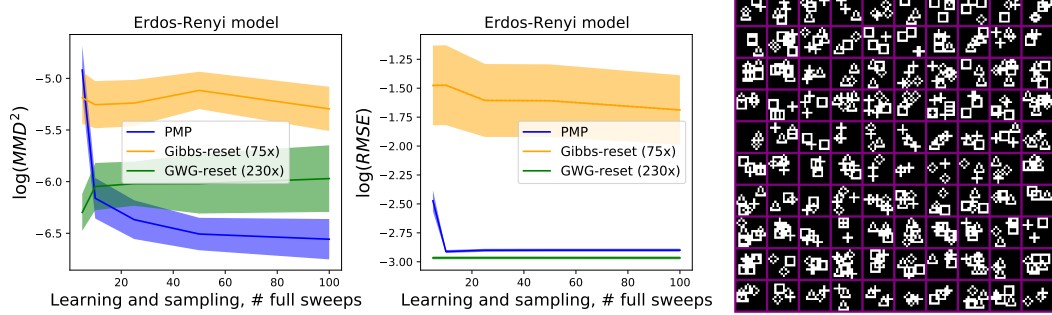

Figure 5: Left and center: Performance on the Erdös-Renyi graphs from [8] for non-persistent training. The x-axis determines the number of full sweeps for training and sampling, for all methods. Right: Full blind deconvolution dataset, 100 examples.

- It consists of 625 binary variables, connected in a periodic, cyclic $25 \times 25$ 2D lattice. The average number of neighbors of a node is also 4.

- The energy expression for this model is $E(x) = x^\top W x$, where $W$ is symmetric and has zero diagonal. The binary variables $x$ are in $\{-1, +1\}^{625}$. Note that the customary 0.5 factor on the pairwise weights is missing, so each weight is counted twice.

- When two nodes are connected, the weight of their connection is set to $\theta$ (fixed to $-0.1$ in this work). However, that weight will be counted twice, see previous item.

- There is no bias term in this model.

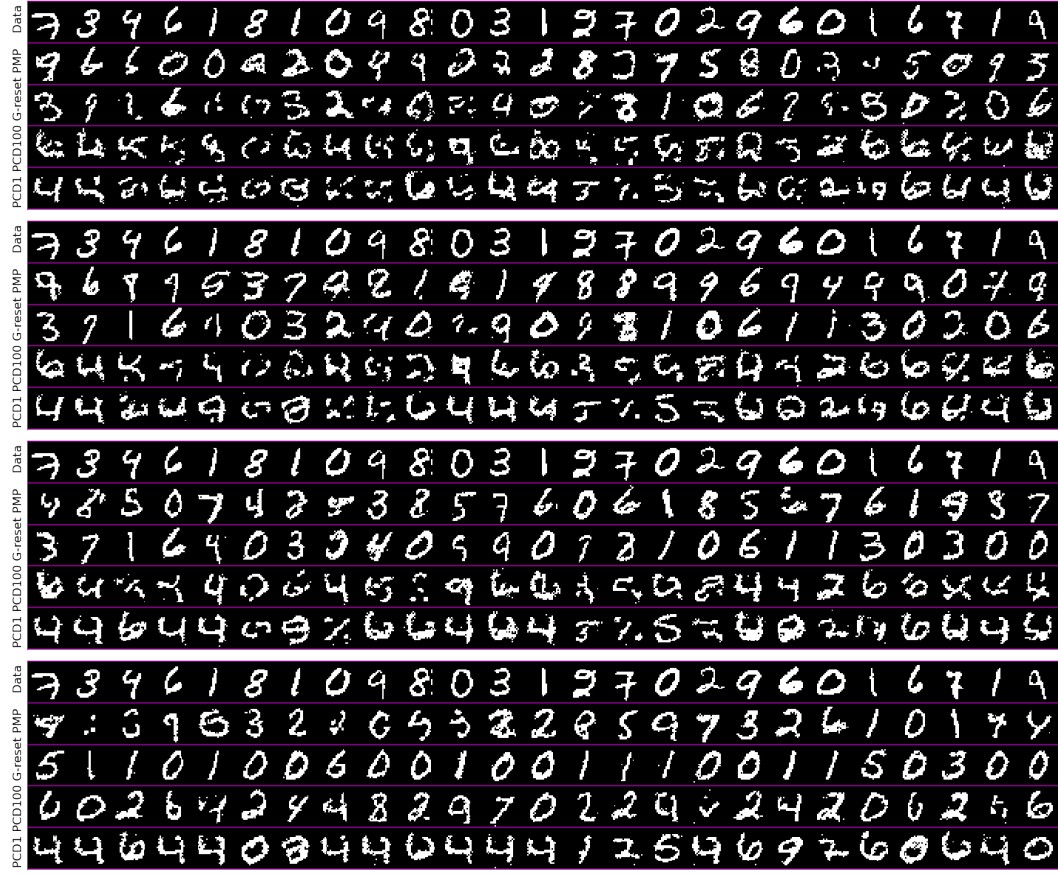

Figure 6: Data and 25 samples per method at $\{50, 100, 200, 2000\}$ full sweeps (one per block).

For both models, the dataset is generated by running Gibbs sampling for 1,000,000 iterations, 2,000 times (thus generating 2,000 samples). The non-persistent training uses 1,000 learning iterations, with as many full sweeps as specified in between those. The persistent training follows [8] and uses 100,000 individual variable updates (or equivalent cost for PMP). We use Adam with learning rate 0.001 and regularize the weights with an $\ell 1$ penalty with strength 0.01. The structure of the model is not used for training, i.e., a completely general Ising model is learned. The root-mean-square error is computed between the ground truth matrix $W$ and its estimation, considering all its entries (including the diagonal, which will be zero in both cases). The bias term is ignored.

We also provide the results with non-persistent training for the Erdös-Renyi graph in Figure 5[left and center], analogous to the ones provided in Figure 2[left] for the cyclic 2D lattice.

## B.2    Additional samples from the RBM experiments in Section 5.5

In the main paper, we provide 10 samples after 100 full sweeps for all the considered sampling methods. It is interesting to see the visual quality of the samples after different numbers of full sweeps, so we provide 25 samples at multiple numbers of full sweeps in Figure 6.

Note that visual quality can be misleading. For instance, when looking at the PCD1-trained model sampled for 2,000 full sweeps of Gibbs sampling (last block, last row), the subjective quality of each individual sampled digit might not be stellar, but it is acceptable. However, the performance as measured by MMD is dismal. This might seem counterintuitive until we realize the lack of diversity in the sampled digits, with a strong bias towards a particular version of digit 4. Even if the 4 is acceptable, its relative weight is way too high, thus explaining the poor density match measured by MMD.

## B.3    RBM samples on a single digit: comparison with LP solver

We propose an additional experiment where we train an RBM with 250 hidden variables on the 2s on the MNIST dataset and compare PMP and other methods with the LP solver. Because the LP solver is prohibitively slow, we only train for 1000 iterations, using Adam with learning rate 0.01. We use an Amazon Web Service c5d.24xlarge instance with 96 CPUs for LP training. Every iteration consider 50 chains, and we use 100 sweeps for the different sampling methods. We report the MMDs and present samples from the different methods in Figure 7.

PMP still achieves the best MMD and produces a rich variety of samples, with different shapes as in the MNIST dataset. The LP solver is the second best method in terms of MMD. It is however three orders of magnitude slower than PMP, and does not capture the same variety of samples.

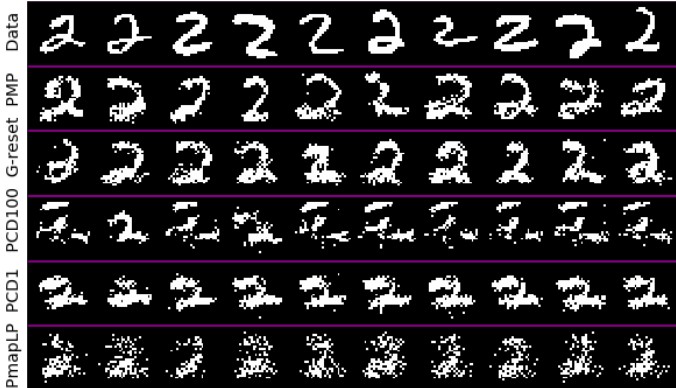

| Method | MMD |
|---|---|
| PMP | **-5.36** (0.06) |
| Gibbs-reset | -4.18 (0.05) |
| PDC100 | -2.70 (0.22) |
| PDC1 | -2.88 (0.22) |
| PmapLP | -4.38 (0.27) |

Figure 7: [Left] RBM samples for the different methods after training on 2s only for 1000 iterations. [Right] Corresponding MMD: we use 100 full sweeps for the sampling methods.

## B.4 About previous work on perturb-and-MAP training of RBMs

[32] presents a Perturb-and-MAP-based method for RBM learning. It is also used to learn an RBM with 500 hidden units on MNIST digits. The authors report success by providing Annealed Importance Sampling (AIS) figures that are very similar to those obtained with contrastive divergence. They however do not show any samples from their trained RBMs.

Since the authors released their code, we ran it (with their chosen parameters) and tried to sample from the resulting RBM weights. We tried Gibbs sampling (which makes sense if AIS is a valid way to measure the quality of the model), PMP (which makes sense if the weights are to be interpreted as valid for perturb-and-MAP sampling) and coordinate descent (the method proposed by the authors). We tried 100 and 10,000 steps for each of these methods. The same approach is applied to the RBM we trained in Section 5.5. Results are displayed on Figure 8. For coordinate descent samples very few pixels turned on, so we did not include them.

We were not able to obtain any meaningful MNIST samples from the RBM trained according to [32]. A possible explanation is that their method actually did not properly train the RBM, and yet AIS was incorrectly reporting a good figure, since AIS is known to be a (sometimes very) optimistic approach to log-likelihood estimation. This optimism might also be due to inadequate hyperparameters within AIS. We were in fact able to obtain extremely (and incorrectly) good AIS scores when training with PMP. In light of this, we decided to use MMD in this work because it is neither optimistic nor its value depends on other hyperparameters that have to be carefully tuned (such as the ones that control the mixing of AIS).

Based on the above, we believe this to be the first work to successfully train an RBM on the MNIST dataset using perturb-and-MAP, and it is definitely the first one to do that and that can sample recognizable, MNIST-like digits using perturb-and-MAP.

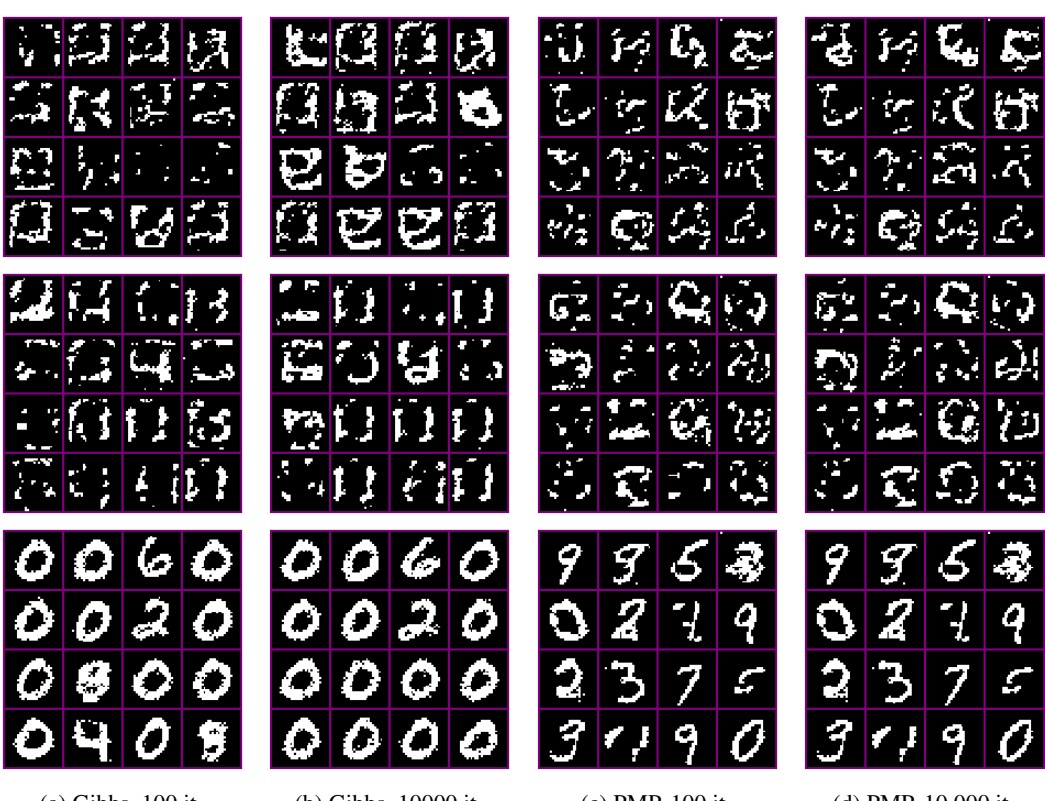

(a) Gibbs, 100 it.  (b) Gibbs, 10000 it.  (c) PMP, 100 it.  (d) PMP, 10,000 it.

Figure 8: [First row] Trained with perturb and coordinate descent, 1 step. [Second row] Trained with perturb and coordinate descent, 10 step. [Third row] Trained with PMP, 100 full sweeps.

# C   Max-product updates for Ising models, RBM, `OR` and `AND` factors

We derive efficient and numerically robust max-product update equations for Ising and RBM models, as well as for the `OR` and `AND` factors used in the Section 5.6.

## C.1   Ising models

We consider a binary Ising model over $n$ binary variables with energy:

$$E(x) = -\frac{1}{2} x^T W x - b^T x, \ \ \forall x \in \{0,1\}^n. \tag{5}$$

This energy defines a probability $p(x) = \frac{1}{Z} e^{-E(x)}$, where $Z$ is the partition function. By factorizing this probability over the nodes and edges of the Ising graph, we observe that two different potentials are involved: (a) joint potentials $\phi_{ij}(x_i, x_j) = \exp(w_{ij} x_i x_j) \ \ \forall i,j$ and (b) unaries potentials $\theta_i(x_i) = \exp(b_i x_i) \ \ \forall i$. It then holds:

$$p(x) = \frac{1}{Z} \prod_i \theta_i(x_i) \prod_{i \neq j} \phi_{ij}(x_i, x_j).$$

**BP updates:**   The max-product update [3] for the message going from the $i$th to the $j$th variable (only defined for $j \neq i$) is defined as:

$$m_{i \to j}(x_j) \ \propto \ \max_{x_i \in \{0,1\}} \left\{ \theta_i(x_i) \phi_{ij}(x_i, x_j) \prod_{k \neq j} m_{k \to i}(x_i) \right\}, \ \ \forall x_j \in \{0,1\},$$

which is equivalent to saying that

$$m_{i \to j}(x_j) \ \propto \max \left\{ \theta_i(0) \phi_{ij}(0, x_j) \prod_{k \neq j} m_{k \to i}(0), \ \ \theta_i(1) \phi_{ij}(1, x_j) \prod_{k \neq j} m_{k \to i}(1) \right\}.$$

After running max-product for a fixed number of iterations, the beliefs are computed via:

$$b(x_i) \ \propto \theta_i(x_i) \prod_k m_{k \to i}(x_i),$$

and we assign each variable to the value with larger belief.

For numerical stability, we first normalize the messages such that:

$$\max(m_{i \to j}(0), \ \ m_{i \to j}(1)) = 1, \ \ \forall i,j.$$

Second, we map messages to the log-space and observe that, because of the above normalization, both $m_{i \to j}(0)$ and $m_{i \to j}(1)$ can be derived from a single message $n_{i \to j}$ defined as:

$$n_{i \to j} := \log(m_{i \to j}(1)) - \log(m_{i \to j}(0))$$

$$= \max \left\{ \sum_{k \neq j} \log m_{k \to i}(0), \ \ b_i + w_{ij} + \sum_{k \neq j} \log m_{k \to i}(1) \right\}$$

$$- \max \left\{ \sum_{k \neq j} \log m_{k \to i}(0), \ \ b_i + \sum_{k \neq j} \log m_{k \to i}(1) \right\}$$

$$= \max \left\{ 0, \ \ b_i + w_{ij} + \sum_{k \neq j} (\log m_{k \to i}(1) - \log m_{k \to i}(0)) \right\} \tag{6}$$

$$- \max \left\{ 0, \ \ b_i + \sum_{k \neq j} (\log m_{k \to i}(1) - \log m_{k \to i}(0)) \right\}$$

$$= \max \left\{ 0, \ \ b_i + w_{ij} + \sum_{k \neq j} n_{k \to i} \right\} - \max \left\{ 0, \ \ b_i + \sum_{k \neq j} n_{k \to i} \right\}.$$

In addition, the MAP assignment is derived as follows:

$$x_i = 1 \text{ if } b_i + \sum_{k \neq i} n_{k \to i} \geq 0 \text{ and } x_i = 0 \text{ o.w.}$$

For computational efficiency we define $N \in \mathbb{R}^{n \times n}$ such that $N_{ij} = n_{i \to j}$ with $N_{ii} = 0$ and we observe that Eq. (6) can be expressed as:

$$P = N^T 1_n 1_n^T - N^T + b 1_n^T$$
$$N = \max(0_{n \times n}, \ P + W) - \max(0_{n \times n}, \ P).$$

where $1_n \in \mathbb{R}^{n \times 1}$ and $0_{n \times n} \in \mathbb{R}^{n \times n}$ are a vector (resp. a matrix) of 1s (resp. 0).

## C.2  Restricted Boltzmann machine

The energy of a binary RBM is $E(v, h) = -h^T W v - b^T h - c^T v$. The max-product updates in log-space can be derived similarly to the above. We now define $n_{HV}$ (resp. $n_{VH}$) the messages going from the hidden variables to the visible ones (resp. from the visible to the hidden). We obtain:

$$n_{i \to j}^{HV} = \max\left\{0, \ b_i + w_{ij} + \sum_{k \neq j} n_{k \to i}^{VH}\right\} - \max\left\{0, \ b_i + \sum_{k \neq j} n_{k \to i}^{VH}\right\}$$

## C.3  Max-product updates for `OR` and `AND` factors

We finally consider the `OR` and `AND` factors involved in Section 5.6. The bottom variable of an `OR` factor is 1 if and only if at least one variable is 1. The bottom variable of an `AND` factor is 1 if and only if all the variable are 1. In our experiments we only consider `AND` factor with two top-level variables.

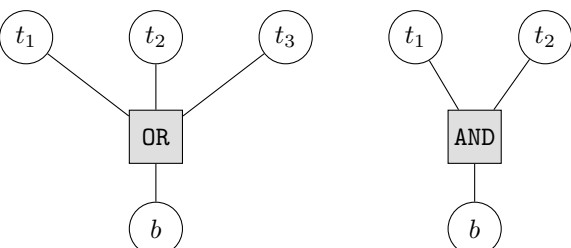

Figure 9: Factor graph for an `OR` factor [left] and an `AND` factor with two top-level variables [right].

**Notations:**  We note $m_{f \to c}(x_c) > 0$ (resp. $m_{c \to f}(x_c) > 0$ ) the message in real space going from a factor $f$ to a binary variable $c$ with value $x_c \in \{0, 1\}$ (resp. from a variable $c$ with value $x_c$ to a factor $f$). We map messages to log space by defining $n_{f \to c} = \log(m_{f \to c}(1)) - \log(m_{f \to c}(0))$.

`AND` **factor, forward pass:**  We derive herein the expression of the message going from the `AND` factor to the top variable $t_1$. By definition it holds

$$\begin{cases} m_{f \to t_1}(1) = \max\{m_{b \to f}(1) \, m_{t_2 \to f}(1), \ m_{b \to f}(0) \, m_{t_2 \to f}(0)\} \\ m_{f \to t_1}(0) = \max\{m_{b \to f}(0) \, m_{t_2 \to f}(0), \ m_{b \to f}(0) \, m_{t_2 \to f}(1)\} \end{cases}$$

It consequently holds:

$$\begin{aligned} n_{f \to t_1} &= \max\{\log m_{b \to f}(1) + \log m_{t_2 \to f}(1), \ \log m_{b \to f}(0) + \log m_{t_2 \to f}(0)\} \\ &\quad - \max\{\log m_{b \to f}(0) + \log m_{t_2 \to f}(0), \ \log m_{b \to f}(0) + \log m_{t_2 \to f}(1)\} \\ &= \max\{n_{b \to f} + n_{t_2 \to f}, 0\} - \max\{n_{t_2 \to f}, 0\} \end{aligned}$$

A similar expression hols for $n_{f \to t_2}$.

**AND factor, backward pass:** We consider herein the message going from the AND factor to the bottom variable $b$. By definition, we have:

$$\begin{cases} m_{f\to b}(1) = m_{t_1\to f}(1)\, m_{t_2\to f}(1) \\ m_{f\to b}(0) = \max\{m_{t_1\to f}(0)\, m_{t_2\to f}(0),\ m_{t_1\to f}(0)\, m_{t_2\to f}(1),\ m_{t_1\to f}(1)\, m_{t_2\to f}(0)\} \end{cases}$$

It consequently holds:

$$n_{f\to b} = \min\{n_{t_1\to f} + n_{t_2\to f},\ n_{t_1\to f},\ n_{t_2\to f}\}$$

**OR factor, forward pass:** We now consider the messages going from the OR factor to the top variables. To this end, we need to consider the variable with the largest incoming message separately. Let us assume that $m_{t_1\to f} = \max_i\{m_{t_i\to f}\}$ and that $m_{t_2\to f} = \max_{i>1}\{m_{t_i\to f}\}$. For $i > 1$ we have

$$\begin{cases} m_{f\to t_i}(1) = m_{b\to f}(1) \displaystyle\prod_{j\neq i} \max\{m_{t_j\to f}(1),\ m_{t_j\to f}(0)\} \\[2mm] m_{f\to t_i}(0) = \max\left\{ m_{b\to f}(0) \displaystyle\prod_{j\neq i} m_{t_j\to f}(0),\ \ m_{b\to f}(1)\, m_{t_1\to f}(1) \displaystyle\prod_{j\neq 1,i} \max\{m_{t_j\to f}(1),\ m_{t_j\to f}(0)\} \right\} \end{cases}$$

It consequently holds:

$$n_{f\to t_i} = n_{b\to f} + \sum_{j\neq i} \max\{0,\ n_{t_i\to f}\} - \max\left\{0,\ n_{b\to f} + n_{t_1\to f} + \sum_{j\neq 1,i} \max\{0,\ n_{t_i\to f}\right\}$$

$$= \min\left\{ n_{b\to f} + \sum_{j\neq i} \max\{0,\ n_{t_i\to f}\},\ \ \max\{0,\ n_{t_1\to f}\} - n_{t_1\to f} \right\}.$$

A similar reasoning for $i = 1$ gives:

$$n_{f\to t_1} = \min\left\{ n_{b\to f} + \sum_{j\neq 1} \max\{0,\ n_{t_i\to f}\},\ \ \max\{0,\ n_{t_2\to f}\} - n_{t_2\to f} \right\}.$$

**OR factor, backward pass:** We finally derive an expression for the message going from the OR factor to the bottom variable $b$. We also assume that $m_{t_1\to f} = \max_i\{m_{t_i\to f}\}$. It then holds:

$$\begin{cases} m_{f\to b}(1) = m_{t_1\to f}(1) \displaystyle\prod_{j\neq 1} \max\{m_{t_j\to f}(1),\ m_{t_j\to f}(0)\} \\[2mm] m_{f\to b}(0) = \displaystyle\prod_j m_{t_j\to f}(0), \end{cases}$$

from which we derive

$$n_{f\to b} = n_{t_1\to b} + \sum_{j\neq 1} \max\{0,\ n_{t_i\to f}\}.$$

## D  LP relaxations for MAP inference

In this section, we derive efficient LP relaxations for MAP inference in Ising and RBM models. The formulations we propose are equivalent to the standard LP relaxation and produce the same exact relaxed solutions for $x$, but does so faster.

### D.1  LP relaxations for Ising models

For a fully connected binary Ising model with energy given by Eq. (5), the MAP problem is defined as:

$$\max_{x\in\{0,1\}^n} \sum_{1\le i\neq j\le n} \frac{1}{2} w_{ij} x_i x_j + \sum_{i=1}^n b_i x_i. \tag{7}$$

We denote by $\{q_{ij}(x_i, x_j) \mid 1 \le i \ne j \le n;\ x_i, x_j \in \{0,1\}\}$ a distribution over the edges of the model and $\{p_i(x_i) \mid 1 \le i \le n;\ x_i \in \{0,1\}\}$ a distribution over the nodes. The standard LP relaxation [28] of Problem (7) is:

$$\max_{p,q} \sum_{1 \le i \ne j \le n} \sum_{0 \le x_i, x_j \le 1} \tfrac{1}{2} w_{ij} q_{ij}(x_i, x_j) + \sum_{i=1}^{n} \sum_{0 \le x_i \le 1} b_i p_i(x_i).$$

$$\sum_{0 \le x_i \le 1} p_i(x_i) = 1 \qquad\qquad \forall i$$
$$\sum_{0 \le x_i, x_j \le 1} q_{ij}(x_i, x_j) = 1 \qquad\qquad \forall i \ne j$$
$$p_i(x_i) = \sum_{0 \le x_j \le 1} q_{ij}(x_i, x_j) \qquad\qquad \forall i \ne j$$
$$p_j(x_j) = \sum_{0 \le x_i \le 1} q_{ij}(x_i, x_j) \qquad\qquad \forall i \ne j$$
$$0 \le p \le 1, \quad 0 \le q \le 1.$$

The above LP involves $2n + 4n(n-1) = 4n^2 - 2n$ variables and $9n^2 - 6n$ constrains. We propose to solve the smaller and equivalent LP with $n^2$ variables $\{\tilde{p}_i\}_{1 \le i \le n} \cup \{\tilde{q}_{ij}\}_{1 \le i \ne j \le n}$ and $4n^2 - 3n$ constrains defined as:

$$\max_{\tilde{p}, \tilde{q}} \sum_{1 \le i \ne j \le n} w_{ij} \tilde{q}_{ij} + \sum_{i=1}^{n} b_i \tilde{p}_i.$$

$$\tilde{q}_{ij} \le \tilde{p}_i \qquad\qquad \forall i \ne j$$
$$\tilde{q}_{ij} \le \tilde{p}_j \qquad\qquad \forall i \ne j$$
$$\tilde{p}_i + \tilde{p}_j - 1 \le \tilde{q}_{ij} \qquad\qquad \forall i \ne j$$
$$\tilde{p} \le 1, \ 0 \le \tilde{q}.$$

**Proof:** To prove the equivalence, start from a feasible solution $(\tilde{p}, \tilde{q})$ of the second problem and define $p_i(1) = \tilde{p}_i$, $p_i(0) = 1 - \tilde{p}_i$ and $q_{ij}(1,1) = \tilde{q}_{ij}$, $q_{ij}(1,0) = \tilde{p}_i - \tilde{q}_{ij}$, $q_{ij}(0,1) = \tilde{p}_j - \tilde{q}_{ij}$, $q_{ij}(0,0) = 1 - \tilde{p}_i - \tilde{p}_j + \tilde{q}_{ij}$ and observe that we obtain a feasible solution $(p, q)$ of the first problem with same objective value.

### D.2 LP relaxations for RBMs

We consider a binary RBM with $m$ hidden and $n$ visible variables. We introduce the variables $\{\tilde{h}_i\}_{1 \le i \le m} \cup \{\tilde{v}_j\}_{1 \le j \le n} \cup \{\tilde{z}_{ij}\}_{1 \le i \le n,\ 1 \le j \le m}$. Similar to above, the LP relaxation of the MAP assignment problem can be expressed as:

$$\max_{h,v,z} \sum_{1 \le i \ne j \le n} w_{ij} \tilde{z}_{ij} + \sum_{i=1}^{n} b_i \tilde{h}_i + \sum_{j=1}^{n} c_j \tilde{v}_j$$

$$\tilde{z}_{ij} \le \tilde{h}_i \qquad\qquad \forall i \ne j$$
$$\tilde{z}_{ij} \le \tilde{v}_j \qquad\qquad \forall i \ne j$$
$$\tilde{h}_i + \tilde{v}_j - 1 \le \tilde{z}_{ij} \qquad\qquad \forall i \ne j$$
$$\tilde{h} \le 1, \ \tilde{v} \le 1, \ 0 \le \tilde{z}.$$