# OpenReview forum: "Perturb-and-max-product: Sampling and learning in discrete energy-based models"
_NeurIPS.cc/2021/Conference — NeurIPS 2021 Poster_

### Official Review · Reviewer_hSgj · 2021-07-15

**Rating:** 7
**Confidence:** 3

**Summary:**

Perturb & Max is an established method for sampling (and therefore learning) a certain class of energy-based models, namely latent-variable models whose joint distribution over visibles & latents is exponential family. However, the max operation poses computational difficulties and existing approaches based on LP solvers are rather limited. This paper proposes an alternative approach, called PMP, that is based on max-product messaging passing. PMP shows favourable empirical performance compared to LP-based solvers, as well as a recent state-of-the-art Gibbs-with-Gradients.


**Limitations And Societal Impact:**

I have made recommendations regarding technical limitations above. I am happy with the societal impact section.

**Main Review:**

**Overall verdict**

This paper was well-communicated, had a couple of surprising insights and strong empirical results. I enjoyed reading it, and recommend it for acceptance. My only complaint is that I felt that certain limitations of the method (and advantages of competing methods) were understated.

**Originality & Quality**

PROS

1) One of the author's key observations is that the LP & max-product solutions to Equation 4 yield different results. This is an interesting observation and, crucially, enables the authors to improve upon existing LP solvers.

2) The exprimental results are generally strong. In particular, the RBM results are quite remarkable: PMP outperforms block-gibbs (visually & in terms of MMD) *without* knowledge of the model's conditional independencies (although you do pay for it in terms of computational cost).

3) The parallelizability of PMP is very nice, and its good to see that there is a JAX implementation.

4) Whilst experiment 5.6 was not easy to understand without delving into the appendix, I apppreciated that the authors had gone to the effort of constructing a high-dim complex task which appears to stretch the limits of state-of-the-art samplers like GWG. I expect this experimental setup could be useful in future work for testing the efficacy of novel discrete samplers.

CONS

1) The authors claim that the "[energy-based] models can be arbitrary as long as they are built using tractable factors", and their method handles "black-box parameter estimation for a very general class of models". Whilst neither claim is false, I think they give a misleading impression. I would prefer more direct statements of the form: "We assume a joint distribution, over observed and latent variables, is exponential family (i.e. the energy is linear in the parameters)". Whilst such models are very useful and popular, they do have limitations. For instance, neural energy functions are not included, whereas competing methods such as Gibbs & Gibbs-with-Gradients are applicable here - see Section 8 in [1].

2) In fact, there are even greater parameteric restrictions than the previous point suggests. The authors say in line 161 that the factors in the model must be "max-marginalizable". I wasn't sure exactly what was meant by this phrase (please clarify) but I inferred it to mean that the message-passing updates are tractable to compute. This does seem like a fairly big limitation.

2) The question of which perturbation distribution to use seems pretty crucial, as it controls the amount of bias in PMP. I would have liked to have seen experiments that varied the choice of perturbation distribution (and the parameters of these distributions) to see how much impact it has.

3) I would have liked to see other metrics beyond MMD. Annealed Importance Sampling can be used to approximate the partition function and therefore the average log-likelihood (and for small RBMs, exact computation is feasible).

4) As explained in Section 5.1, the parameters estimated by PMP will not, in general, match ground truth parameters (even for identifiable models). This implication is that PMP isn't exactly a method for learning energy-based models, but rather a method for learning a sampler with an EBM-based parameterisation. This stands in tension with the claim (in the title & abstract) that PMP is a method for sampling *and* learning of EBMS. That said, the RMSE results in Figure 2 do at least provide some empirical support to the claim that PMP can accurately learn the energy function. I would be even more convinced if there were AIS calculations for the RBM experiments, as this would shed some more light on how well the energy function has been estimated.

**Clarity**

In section 2.3, the author's take an 'apparent detour'. I don't like this writing style; it requires the readers to hold too much in memory, and then re-read parts of section 2.2 after finishing 2.4 in order to make sense of everything. In particular, it should be possible to introduce Equation 4 in a way that directly relates to Equation 3.

Equation 1 (and its gradient in line 57) include latent variables. But Equation 3 and its gradient in line 80 do not. I think Equation 3 should be altered to include latent variables, since this would help in understanding all the steps in Algorithm 1.

**Significance**

I think this paper has a decent chance of being widely read and cited in the EBM literature. Moreover, many communities outside of ML use energy-based models (e.g. Potts models in proteomics) and the algorithm presented here may be very useful to them.

**References**

[1] Grathwohl, Will, et al. "Oops I Took A Gradient: Scalable Sampling for Discrete Distributions." arXiv preprint arXiv:2102.04509 (2021).

**minor points/questions**

- The expression for the energy in line 50 looks incorrect. Where are the parameters \theta_i?
- Could PMP be tractably applied to deep Boltzmann machines?


**Time Spent Reviewing:**

4.5

---

> ### Author Response · Authors · 2021-08-10
> **Author response**
>
> Thank you for your careful review! In light of your comments we will make several modifications to improve the quality of the manuscript:
>
> - We will emphasize that neural functions are not supported.
>
> - We will clarify that (as you correctly assumed) max-marginalizable means that the message passing updates are tractable. However, note that for factors with small arity (i.e., pairwise, ternary), these are always tractable. For logic functions such as AND an OR, etc, they are also tractable even if the arity is large. In fact, the only common intractable case that we can think of are neural functions as you pointed out.
>
> - We will clarify the choice of perturbation distribution. Perturb-and-MAP theory suggests that for the case of independent perturbations, the Gumbel distribution is a particularly good choice. We also observed in informal experiments that using other densities (such as a Gaussian) performed worse even when tweaking its variance for best performance.
>
> - We will be clearer about the problems of using AIS to assess model quality. In fact, we did initially use AIS to assess our RBM learning and found overly optimistic results (and overly pessimistic ones when using reverse AIS). Despite trying multiple parameterizations and runtimes to increase the quality of AIS, it seems that the resulting models did not produce tight bounds. This has been a problem even with previously published research relying on AIS, as we discussed in Section A3 of the Appendix.
>
> - We will tie in Section 2.3 better into the flow so that it doesn't feel like a detour. The reviewer is absolutely right that Eqs. 3 and 4 are closely connected.
>
> - We will modify the handling of the latent variables for clarity. We will remove them in the initial presentation and then show how including them only requires a minor modification. The only reason to avoid using them throughout the entire presentation is unnecessary clutter.

---

> > ### Comment · Reviewer_hSgj · 2021-09-11
> > **Thank you for your reponse**
> >
> > Thank you for your clarifications and promised amendments to the text.
> >
> > My opinion of the paper and score remains the same.

---

### Official Review · Reviewer_aLeX · 2021-07-16

**Rating:** 6
**Confidence:** 3

**Summary:**

The manuscript is clear, well written, and presents some interesting results in the use
of the Belief Propagation (BP) algorithm to train discrete energy-based models.

The authors investigate the use of BP in its zero-temperature/max-product version to
approximately solve the MAP phase of the Perturb-and-MAP approach, as opposed to costly and
less effective linear programming relaxations or other heuristics.

**Limitations And Societal Impact:**


In my opinion, the major limitations of the paper are the following:

- The approach is not suitable for energy models that are not highly factorized,
  as in the notable case of an energy function expressed by a multi-layer perceptron.

- On loopy graphical models, BP is an uncontrolled approximation. Comfortably though,
  their experiments on loopy systems show good results.

- The framework doesn't allow the learning of an explicit energy model, but of an implicit sampler where the epsilon noise distribution plays an important role. Therefore, the possibility of composing energies, one of the few advantages of EBM over other generative models are lost. Also, it cannot be used to solve inverse problems (i.e. estimate the energy parameters).

The paper marks good progress in the field of discrete energy-based models and in
the application of message passing algorithms, I'm not sure it's up to the NeurIPS
innovation standards though.

Minor comments:

- Some of the hyperparameters used in the experiments are not entirely clear. E.g. from line 229
  "100 chains, 50 full sweeps" I infer that the perturbation epsilon is sampled 100 times,
  and each time the algorithm is run for 50 iterations to find the approximate MAP. This could be clarified a bit. Also, I could not find the number of gradient steps or epochs used.

- The definition of MMD could be reported to facilitate the reading.

- Fig. 2: not clear how Zapprox is computed

**Main Review:**

Relations to works in the area are carefully discussed. The authors provide a good coverage
of experiments and datasets, and compare to standard Gibbs sampling and to the recently introduced
Gibbs-with-gradients.
Unfortunately, there is little novelty in the technique proposed, but the surprisingly good performance of BP in the proposed approach is a nice discovery.



**Time Spent Reviewing:**

3

---

> ### Author Response · Authors · 2021-08-10
> **Author response**
>
> Thank you for your review! Let us address each of your points separately below.
>
> - About the major limitations of the paper
>
> We very much agree with the first two limitations that you mention (unsuitable for non-factorized models such as NNs, and loopy BP being an uncontrolled approximation). We also agree that this method cannot be used to solve inverse problems (precisely estimate the Gibbs $\theta$​). We will emphasize in the paper that multilayer perceptrons are not supported.
>
> We disagree with the third limitation (lack of composability of the energies), depending on how it is interpreted. Composing energies is perfectly possible in our proposal as long as we train the system as a whole. This doesn't negate the main advantage of composability, which is a modular design, or even the testing of individual pieces, but it does require fine-tuning of the joint system. As far as we understand, this is always true: if we have a loopy graphical model composed of pairwise factors (energy pieces), we can never expect to train those pairwise factors separately (even if we did it exactly), compose them, and obtain the best fit to data without further joint fine-tuning. The noise source is always Gumbel noise, so this is not a factor that varies from piece to piece either.
>
> In the sense of composability, this approach is identical to any other: it provides an approximate fit for the pieces, and when the pieces are composed, the global model is roughly trained and might work in practice, but for best fit requires joint fine-tuning. The experiments show that the RMSE over $\theta$ is not terribly bad either.
>
> - Some of the hyperparameters used in the experiments are not entirely clear. E.g. from line 229 "100 chains, 50 full sweeps" I infer that the perturbation epsilon is sampled 100 times, and each time the algorithm is run for 50 iterations to find the approximate MAP. This could be clarified a bit. Also, I could not find the number of gradient steps or epochs used.
>
> Your understanding is correct. The number of gradient steps is specified in the previous line "we use ADAM for 1000 iterations". We use all samples at each gradient step, so the number of steps and epochs is the same, 1000. We will clarify this.
>
> - The definition of MMD could be reported to facilitate the reading.
>
> We will add the definition in the supplementary
>
> - Fig. 2: not clear how Zapprox is computed
>
> The partition function does not need to be approximated to compute either the MMD (only samples are necessary) or RMSE (only an estimation of the parameters $\Theta$ is necessary), so it is not computed.

---

> > ### Comment · Reviewer_aLeX · 2021-08-26
> > **comment**
> >
> > > We disagree with the third limitation (lack of composability of the energies), depending on how it is interpreted. Composing energies is perfectly possible in our proposal as long as we train the system as a whole. This doesn't negate the main advantage of composability, which is a modular design, or even the testing of individual pieces, but it does require fine-tuning of the joint system. As far as we understand, this is always true: if we have a loopy graphical model composed of pairwise factors (energy pieces), we can never expect to train those pairwise factors separately (even if we did it exactly), compose them, and obtain the best fit to data without further joint fine-tuning. The noise source is always Gumbel noise, so this is not a factor that varies from piece to piece either.
> >
> > I agree with the author's reply

---

### Official Review · Reviewer_MkBQ · 2021-07-16

**Rating:** 6
**Confidence:** 4

**Summary:**

This paper proposes to use max-product belief propagation to provide an approximate solution to the MAP problem encountered in the perturb-and-MAP sampling method. The paper argues that, despite its lack of convergence guarantees, max-product is faster (due in large part to its parallelizability) and better for sampling than linear-programming based methods (the standard approach) in problems with multiple modes. Experiments in a number of synthetic tasks show that the proposed approach, perturb-and-max-product (PMP), is an efficient and effective sampling method that can also be used for parameter estimation of discrete energy-based models in a black-box manner.

**Limitations And Societal Impact:**

Yes, the discussion in section 3 and elsewhere provides a nice summary of the limitations and tradeoffs of the method. Societal impact is not discussed but the presented method does not require such a discussion.

**Main Review:**

# Overall
In summary, I found this paper to be a straightforward and pleasant read. The paper very clearly lays out the problem being tackled, the proposed approach, and how this proposed approach fits in to existing work in this area. The proposed method is, to my knowledge, novel. I was somewhat surprised by this, as using max-product for perturb-and-MAP seems like something that would have been tried before; however, I do not know of and did not find existing work taking this approach so it indeed seems novel, and the fact that the method appears obvious in hindsight takes nothing away from the paper. More importantly, the method outperforms the compared baselines quite consistently. While the experimental domains are all synthetic and straightforward, they are common domains for this problem and do demonstrate the benefits of the method, both for sampling and parameter estimation. That said, there are some minor issues with clarity and explanation of one of the experiments (blind deconvolution), and a fairly major issue with a missing baseline in the experiments, making the comparisons seem unfair.

# Quality
The submission is sound and claims are mostly well-supported (see below). The paper does a great job of justifying the decisions made throughout and providing supporting evidence for them. The authors are careful and honest about the method and its evaluation in relation to what is contained in the paper. I enjoyed reading the paper until I reached the experimental section where it seems the most natural PMP baseline---that of perturb-and-MAP with a LP solver---is not included (indeed, this is even the standard method reference in the abstract). This seems like a fairly significant issue with the experiments, and I do not see why it would not be included. Even if it takes much longer to run, then run it for a reasonable amount of time and show the results. I am aware that the authors do compare to MPLP in section 5.2 and state that it is too slow for sampling, but it could presumably at least be used for 1-10 sweeps (instead of 200) to get the timescale down to a reasonable amount.

Beyond this, there are two other areas where the authors could further strengthen the paper. The first would be to provide a better understanding and quantification of the relationship between the learned parameters $\Theta_{PMP}$ and the parameters of the Gibbs distribution (as discussed in Section 5.1), and what role, if any, this plays when using the learned model. The second would be to show the benefit of PMP in a more challenging experiment on real-world data. I do not think that either of these are necessary but do think their inclusion would strengthen the paper.


# Clarity
As stated above, the paper is overall quite clear, well-organized, and informative to the reader. However, one area in which I found the paper confusing and had to re-read sections to properly understand is in relation to the hidden variables $x_h$. These are introduced in eq (1) clearly enough but then disappear in the approximation, eq (3). When reading through, I interpreted this as implying that the remainder of the paper would not concern itself with hidden variables. But then the update rules in Algorithm 1 and the text on lines 146-147 indicate that the observations $x^{(n)}$ are in fact partial and thus there are hidden variables but the hidden variables are never mentioned explicitly after eq (1). I would encourage the authors to be more consistent and specific in their use of the hidden variables in eq (3) and beyond, and to mention them explicitly in the text beyond section 2.1.

The second area that I found difficult to understand was the exact setup of the blind deconvolution experiment. Some of my questions about this section are as follows
1. It's unclear to me what the effect of adding a feature and "making them all bigger" (lines 300-302) for W when generating samples is.
2. Why not sample W and S jointly and produce samples that way?
3. Also if W is being provided for generation, how is this different than the case with GWG discussed on line 306?
4. What is a "reasonable time"? (line 305) What do samples look like if run for an "unreasonable" amount of time?
Overall, I think the presentation of this experiment could be greatly clarified in the main body and expanded if necessary in the supplement (e.g., even if it's a very simple generative model, please write it down for reference). Finally, this experiment shows that PMP can recover some of the shapes in the dataset but does not show that the samples match the dataset at all (images of single larger shapes vs of multiple smaller overlapping shapes). Is this the goal? Unclear from the text.

Other minor issues with clarity:
- Line 43 references "ML learning" without defining the acronym.
- Related to the hidden variables $x_h$, the definition of the dataset on line 52 does not admit the inclusion of missing values unless $D < |\text{domain}(x)|$ unless ${0,1}^D \subset \text{domain}(x)$, but then domain($x$) is not defined.
- The notation $ \Theta {|{x^{(n)}}} $ to represent clamping of variables is confusing, because $\Theta$ is the set of parameters, not a function of the variables. If I understand the notation correctly, this should instead be $\Phi|_{x^{(n)}}$ since $\Phi$ is a function of $x$ and $\Theta$ is not.
- Can you speak to the poor performance of PMP on MMD for the Erdos-Renyi model in Fig 2?

# Significance
The method is novel and interesting and I think will prove useful for other researchers but issues with the experiments hold the paper back.

----------------------------------------------------
### After author response

Thank you for your thorough response. I'm satisfied with the response (as much as possible without seeing an updated version of the paper and results) and have increased my score above the acceptance threshold. I think the paper will be much stronger after the information provided in the author response has been included.

**Time Spent Reviewing:**

5

---

> ### Author Response · Authors · 2021-08-10
> **Author response**
>
> Thanks for your detailed review! Let us address each of your points:
>
> - The experiments section does not contain experiments with PMP + LP solver
>
> We agree that adding this to the supplementary material will highlight the problems with the LP approach further. We have indeed performed these experiments internally, but the results were so bad (both in terms of computational cost and quality of results), that we didn't see this approach even as a meaningful competing algorithm. The problem here is that for any non-attractive model, the LP solution is so far from performing an $\arg\max$​​​​ that the approach completely fails to sample and learn.
>
> One option is to use an off-the-shelf solver and run it to completion. This option is impractically slow, but we agree it will be useful to show that the approach does not learn anything meaningful. We can show, for instance, how sampling from a network trained in this way leads to non-sensical samples.
>
> Another option is (as you suggest) to limit the computational budget for the LP solver. The results here would depend on the specific strategy employed by the solver, so we'd have to narrow it down to a concrete approach. As you suggest, we can use MPLP with fewer iterations. In this case, you will see that with unlimited iterations, the results in Section 5.2 carry over to sampling and learning, and with limited iterations the results are even worse.
>
> We will include both options as additional experiments in the supplementary material, to clearly justify why this approach was not included in the main paper, and summarize them in the main paper.
>
> - The hidden variables are not explicitly included in most equations
>
> Our method does indeed support the use of hidden variables at all points, but explicitly separating visible and hidden variables in all the equations cluttered them, so we decided against it, since no further complication arises from using hidden variables (the same inference mechanism is applied, just twice instead of once). What we will do to make this clearer is to first present the idea with no hidden variables (to avoid clutter in the presentation) and then show how the same approach can be used to solve the case with hidden variables. The description in Algorithm 1 does include the use of hidden variables.
>
> - Lack of clarity in the blind deconvolution experiment
>
> 	- It's unclear to me what the effect of adding a feature and "making them all bigger" (lines 300-302) for W when generating samples is.
>
> 	This is just introducing a mismatch between the generative process and the inference process, to make inference more challenging. The generating feature array W is binary and of size 4x5x5, whereas for inference we set its size to 5x6x6. The extra elements can be set to zero during inference, but having to figure that out makes it more difficult. We do this to show that we don't need to know the size of W precisely, making the deconvolution effectively “more blind”.
>
> 	- Why not sample W and S jointly and produce samples that way?
>
> 	That's exactly what we do. We sample W and S jointly. We only displayed W, which shows that we recovered the features properly. S just tells us where to place those features to reconstruct each image. We didn't include S because W is a better way to visualize if the deconvolution succeeded (if W is correct, obtaining S is much easier, and even GWG would succeed at that). We can indeed show samples from S in the supplementary material.
>
> 	- Also if W is being provided for generation, how is this different than the case with GWG discussed on line 306?
>
> 	The ground truth W and S are provided for the generation of the training dataset. For inference, we only provide X, and sample W and S jointly with no clue about them. GWG can't do that, and only works when S or W is also provided at inference time.
>
> 	- What is a "reasonable time"? (line 305) What do samples look like if run for an "unreasonable" amount of time?
>
> 	With GWG we didn't get results even remotely resembling the true features in hours, whereas PMP solved it in under a minute. Gibbs sampling is asymptotically correct under mild assumptions (that hold here), so running it for an unreasonable amount of time should solve the problem. However, that unreasonable amount of time would grow exponentially with the size of the problem (for problems that are not amenable to Gibbs sampling, like this one).
>
> 	- Finally, this experiment shows that PMP can recover some of the shapes in the dataset but does not show that the samples match the dataset at all (images of single larger shapes vs of multiple smaller overlapping shapes). Is this the goal? Unclear from the text.
>
> 	The convolution operation takes the individual shapes from W and places them in multiple overlapping locations, according to S (which contains the locations). In 1D this is like convolving a waveform W with a signal S that contains several deltas at random locations, multiple overlapping waveforms will result. In the same way (but in 2D), each image is generated by convolving the same set of features W with a new, random, binary, and very sparse S, that selects for each possible location which features (if any) will be placed there. The result that we display is the inferred W, i.e., the individual shapes. We do not display S, which is simultaneously inferred, and which explains how to combine the inferred W to re-create the original images X.
>
> These questions made it clear to us that we need to rewrite this section more clearly, and include samples of S and re-create X in the supplementary, with further explanation.
>
> - Minor questions
> 	- Line 43 references "ML learning" without defining the acronym.
>
> 	Defined in line 22
> 	- Related to the hidden variables xh, the definition of the dataset on line 52 does not admit the inclusion of missing values unless D<|domain(x)| unless 0,1D⊂domain(x), but then domain(x) is not defined.
>
> 	True, will fix this as described above (not using hidden variables, and then extending to use).
> 	- The notation Θ|x(n) to represent clamping of variables is confusing, because Θ is the set of parameters, not a function of the variables. If I understand the notation correctly, this should instead be Φ|x(n) since Φ is a function of x and Θ is not.
>
> 	Agreed, this needs more clarity. In the binary case (for simplicity), for every variable $x_i$​ the energy function contains unary terms of the form $-\theta_0\phi_0(x_i)-\theta_1\phi_1(x_i)$​, where the sufficient statistics $\phi$​ just take the value 0 or 1 as follows $\phi_0(x) = [x=0]$​ and $\phi_1(x) = [x=1]$​​​. Now, to condition the graphical model on the knowledge that, say, $x=1$, I just need to set $\theta_0\rightarrow-\infty$​. In this way, when $x=1$, all energies remain unchanged. And when $x=0$​, all energies are infinity (thus the probability is 0). That's how modifying $\theta$ (and not $\phi$) results in the model being conditioned.
>
> 	- Can you speak to the poor performance of PMP on MMD for the Erdos-Renyi model in Fig 2?
>
> 	Good catch. We noticed this shortly after submission and it is already corrected. In Figure 2, the RMSE and MMD labels are reversed. PMP is performing poorly in terms of RMSE (as expected), not MMD.

---

### Official Review · Reviewer_E7Us · 2021-07-17

**Rating:** 7
**Confidence:** 3

**Summary:**

The paper presents the perturb-and-max-product (PMP) algorithm for sampling and learning in discrete energy-based models (EMB). The algorithm is scalable and parallel. Empirical evaluation is conducted on Ising models, 2D lattices and Erodos-Renyi graphs, and restricted Boltzmann machines. PMP is compared against Gibbs and Gibbs-with-Gradients under various settings.


**Limitations And Societal Impact:**

The limitations are addressed and potential negative societal impact is N/A.


**Main Review:**

I do not have any major comments to make. All-in-all, I think the algorithm is clear, and the paper is well written and executed. The empirical evaluation demonstrates the competitiveness of PMP to the previous mentioned comparable methods.


**Time Spent Reviewing:**

3

---

> ### Author Response · Authors · 2021-08-10
> **Author response**
>
> Thank you for your review, we strived to make the paper complete and easy to read!

---

### Decision · Program_Chairs · 2021-09-27

**Decision:**

Accept (Poster)

**Comment:**

The authors present an interesting framework in which they combine the "perturb-and-MAP" strategy for generating samples from a Gibbs distribution with the perspective that incorrect inference (such as substituting the simple and efficient max-product method) can be compensated by learning the model parameters using a matching approximation, giving a mechanism for (approximately) generating new samples from the training data distribution.

The paper is well-written and clear.  Novelty is somewhat limited, since the work is a straightforward combination of several ideas in the literature.  Experimental validation is reasonable / sufficient, although several reviewers requested some extensions, such as including e.g. off the shelf or limited-budget LP solvers, and expanding on the authors' observations about the performance of such methods.  The authors' response clarified and addressed some of the reviewers' concerns.  Reviewers felt the impact of the work was borderline for NeurIPS, and suggested that some of the authors' claims be toned down or moderated by clear discussion of the method's cons, but were generally in favor of acceptance if possible.

(I also should add that there are *many* method for generating samples from discrete distributions beyond Gibbs-like methods, including tempering methods and "discontinuous" Hamiltonian Monte Carlo for Gibbs distributions, and on the "learning to sample" side, discrete (normalizing) flow models, just to name only a few.)